# Estimating the Internal Oxidation of Ni-Al Alloys

**DOI:** 10.3390/ma13010048

**Published:** 2019-12-20

**Authors:** C.G. Nava-Dino, A. Martinez-Villafañe

**Affiliations:** 1Facultad de Ingeniería, Universidad Autónoma de Chihuahua, Chihuahua, Circuito No 1., Campus Universitario 2 Chihuahua, Chih. C.P. 31125, Mexico; 2Centro de Investigación en Materiales Avanzados (CIMAV), Laboratorio Nacional de Nanotecnología, Miguel de Cervantes No.120, Chihuahua, Chih. C.P. 31109, Mexico; martinez.villafane@cimav.edu.mx

**Keywords:** oxidation, morphology, Fiji software

## Abstract

Ni-Al alloys create a cone- shaped figure when there is internal oxide. This behavior was studied by TEM, SEM, X-Ray (XRD), Optical Microscopy and Image Processing. The internal oxide precipitates and its results indicate that this precipitation forms continuous rods in a cone-shaped configuration extending from the surface to the internal oxide front for Ni-Al alloys, whereas for Ni-X (X = Cr, Mo, V, W and Mn) alloys the precipitation is discrete and more irregularly-shaped. Furthermore, in a high atomic percentage (5.18% to 8.67%), the precipitation was rod-like and continuous from the surface to the internal oxide front for all temperature\time conditions. For the Ni-2.47% Al alloy at 800 °C, observations showed a mixture of rod-like and fork like precipitates, whereas after oxidation at 1000 and 1100 °C the precipitation was rod-like and continuous. For the Ni-1.18% Al alloy the aluminum concentration was insufficient for fully continuous precipitation to develop, and the internal oxides were generally acicular-shaped and discontinuous. Images obtained by TEM and, after that, analyzed by image processing allowed us to understand their behavior and the internal oxide patterns.

## 1. Introduction

Superalloys, such as titanium alloys and nickel based, are generally used in aerospace due to their exclusive combination properties. These alloys are commonly named as difficult-to-cut alloys because of their capacity to preserve their properties at high temperatures which acutely hinders their machinability [1]. Nickel aluminide coatings exhibit high resistance to oxidation, metal dusting and carburization behavior attributed to the formation of alumina protective scales at moderate and high temperature [2]. Internal oxidation is technologically important because the dispersed oxide particles may have an important influence on the properties of the alloy. Ni-base super- alloys are complex engineering alloys, primarily nickel with ten or more alloying elements, designed to operate at temperatures up to 70% of their melting range while resisting mechanical and chemical degradation [3]. Internal oxidation has also been a useful method for measuring the permeability or solubility-diffusivity product of atoms of gases in metals. Ni-based super-alloys containing numerous alloying elements within their microstructures have a significant ability to maintain excellent mechanical performance at elevated service temperatures. Due to their high strength capacity, these materials are substantially used in aerospace (turbine blades/discs), marine and land-based applications. [4,5]. However, from a practical perspective, internal oxidation has usually been considered to be undesirable, since the mechanical properties of structural materials for use at high temperatures are usually degraded by this type of oxidation [6].

The higher oxidation kinetics of the alloys is explained by the presence of oxygen molecules, which penetrate along cracks or fissures through the oxide scales. Many authors were interested in thermal oxidation of nickel which is considered as a model material. In high-purity nickel and in dilute alloys (b5% in weight), it was found that the rate controlling the transport process is the outward diffusion of cations through the Ni_1_−xO scale [7].

In the present research, an attempt has been made to clarify the different morphological aspects of the internal oxidation of several Ni-Al alloys through the characterization by optical, scanning electron and transmission electron microscopy. Moreover, in the typical ways to understand internal oxidation, some studies use prediction models to identify internal oxidation. In some cases fractal patterns in morphology during internal and surface oxidation have been discovered as spheres [8]. In this alloy, a cone shape is observed, and several studies about the morphology of material can be observed and analyzed by a simple and efficient approach for microstructure defect detection based on image analysis. The proposed approach mimics the way that experts identify the defect area, by segmenting material grains via image pre-processing and detecting defects using a region-based graph. The pre-processing step is a sequence of image processing techniques to produce potential defect regions [9]. In this research, layers of morphological aspects were analyzed by Fiji Software^©^.

Ni-based superalloys represent a class of materials designed to withstand extreme conditions; phase is the random substitutional solid solution of Ni, and γ′ phase is the Ni_3_Al based ordered phase with a L1_2_ structure [10].

## 2. Materials and Methods

Samples were made by Ni-1.18% Al, Ni-2.47% Al, Ni-5.18% Al and Ni-8.67% Al in atomic %, using rapid vacuum melting nickel with aluminum of 99.9% purity. These alloys were hot and then cold-rolled to approximately 0.6 mm strip. Specimens (10 × 5 mm) were cut on a 220 grade SiC paper and annealed in an evacuated sealed quartz capsule containing chromium powder as an oxygen getter for 5 h at 1200 °C. The capsules were inserted into a furnace and heated at the reaction temperature (2C) using argon. Observations in the morphology studies of internal oxides precipitates though the internal oxide zone, then DRX (X-ray Diffraction, Malvern Panalytical. Lelyweg, Almelo, The Netherlands) and TEM (transmission electron microscopy, JEOL. Dearborn Road Peabody, MA), were used. After that, images obtained by microscopy were processed by Fiji Software (Laboratory for Optical and Computational Instrumentation (LOCI), University of Wisconsin, Madison, WI, USA), a free open software produced by the US National Institute of Health (NIH).

## 3. Results

### 3.1. Internal Oxidation Depth

The depths of the uniform internal oxidation front from the external surface were measured. Usually, the front was very flat and it was possible to obtain reasonably accurate measurements. When irregularities in the front were observed, the average depth was recorded. Measured depths are plotted in Figure 1, using log-log axes. In the figure, straight lines have been drawn through the points to satisfy the equation 1, with ‘n’ having a value of 0.5.
ξ = Atn(1)
where ξ is the depth of the internal oxidation zone, A is a constant, and t is time and n is exponent time constant. The approximate value of n for an alloy at a given temperature was calculated from the data of only two points; they may not be accurate enough to draw definite conclusions. However, comparison with n values calculated from six different oxidation times at 1000 °C for Ni-1.18%Al and Ni-8.67%Al, giving values of 0.52 and 0.50, respectively, suggests that the values calculated from only two points are reasonable.

#### 3.1.1. Morphology of Internal Oxides.

The internal oxide precipitates developed in the Ni-Al system under study were quite different from those developed for other alloy systems [11,12]. A typical unetched cross-section of a Ni-2.47% Al alloy specimen oxidized at 1100 °C for 10 h is shown in Figure 2a,b, where the internal oxides appeared to be acicular and elongated and were oriented perpendicularly to the specimen surface deeper into the alloy. For the case of Ni-Cr, Ni-Mo, Ni-W and Ni-V alloys, it has been reported that the morphologies of the internal oxide precipitates were small and granular in the vicinity of the specimen surface but become much larger and more irregular in shape with increasing depth from the surface [13,14].

After deep-etching, the same electropolished conditions are used as for the original surface preparation leaving the oxide precipitates. It was noticed that for the Ni-1.18% Al alloy, the aluminum concentration was insufficient for fully continuous precipitates to develop, and the internal oxides were generally acicular but discontinuous (Figure 3a). The morphologies of internal oxidation for the Ni-2.47% Al alloy at 800 °C showed a mixture of rod like and fork precipitates (Figure 3b,c), whereas after oxidation at 1000 and 1100 °C the precipitates were rod-like and continuous, Figure 3d. For the Ni-5.18% Al alloy at 1000 and 1100 °C the morphologies of internal oxidation showed that the precipitates formed were rod-like and continuous from the surface to the internal oxide front (Figure 3e,f). Similar morphologies were also observed for the Ni-8.67% Al alloy at 1000 and 1100 °C (as an example see Figure 7B). Studies for the Ni-Cr system have shown that the internal oxide was never continuous under any temperature/time conditions [15,16]. Instead, the precipitates were discrete, being generally small and granular near the surface but larger and more irregular in shape deeper into the alloy. The specific sizes and shapes of the precipitates were influenced by the temperature, by the nature of the solute element and by its concentration. For instance, in the Ni-Cr system the precipitates tended to be more acicular near the internal oxide/alloy interface and were oriented either perpendicular to the surface or in a relatively random manner with respect to the surface.

#### 3.1.2. Morphology Analysis in Fiji Software

Fiji ImageJ is a common open source image processing software tool based on Java with a long history of scientific image analysis. This cross platform compatible tool enables implementing issue-adapted plugins. Figure 4 shows a FIJI software interface image from Optical microscopy being processed. In the current research, diffusion of oxygen was analyzed by FIJI software using ROI (Region of Interest). This option allows us to select multiple ROI and understand this cross-section zone. Most of the existing research works for automated image analysis of alloys focusing on Metallography Image Analysis (MIA), which is to study the physical properties of metals by applying image analysis methods on microscope images, applied on segmenting the different phases of microstructure. The features are computed in the segmented Regions of Interests (ROIs), commonly using a segmentation algorithm. This algorithm tries to take two vectors of featured values V1 and V2 computed over a set of ROIs (image segments) by two software implementations of the same feature [17]. In Figure 4(1), a yellow line selects ROI; then Figure 4(2) shows a graphical description about the cross section zone where the important changes in morphology regarding to temperature are observed. Figure 4(3) shows the numbers that identify ROI; with this, mathematical analysis was made.

## 4. Discussion

The internal precipitation of alumina, which is of a much higher specific volume than the metal matrix, leads to the formation of high local compressive stress in the vicinity of the precipitates.

When oxygen and Al, both dissolved in nickel, combine to form Al_2_O_3_, there is a considerable increase in volume [18,19]. Unless this volume change is accommodated by mass transfer of nickel atoms, large compressive stress will develop in the matrix around the internal oxide precipitates. If large stress develops, the particles would rapidly grow along the directions in the matrix which allow the smallest rate of increase of stress energy and would not be spherical. Moreover, the stress needs to be relieved or accommodated by shear in the matrix. In the Ni-Al alloy, the very fine distribution of precipitates makes this stress relief difficult. Some relief can be obtained by grain boundary sliding and by flow of alloy adjacent to the grain boundaries. This alloy is denuded of internal oxide precipitates, so it flows rather easily, especially at higher temperatures.

As a matter of fact, it was observed for the Ni-Al alloys that their surfaces were visualized at the alloy grain boundaries. These steps were more evident for Ni-8.67%Al (Figure 2a,b). In these alloys, the steps were always present above a grain boundary intersecting the specimen surface at an oblique angle, and the alloy grain on the side of the sharper angle was always pushed out. The grain on the other obtuse angle side of the grain boundary always remained relatively low. For this grain an internal oxide-denuded zone adjacent to the boundary was observed (Figure 2b). It is possible that such features were caused by the stress created by an increase in volume as the internal oxide was formed in the internal oxidation region.

Observations of internal oxides in the Ni-Al alloys have indicated that the main morphology was granular precipitation for the alloys with small solute contents. This is probably due to the small amount of energy required for the nucleation of a spherical particle. These small granular nuclei can grow in accordance with basic nucleation and grow theory. Subsequent growth occurs only if the radius, r of the nucleus is greater than the critical size, rc, given by the same theory. A stable granular particle growing by coalescence, in an isotropic concentration field, should retain its shape and enlarge until the supply of solute and oxygen is exhausted. In general, the internal oxidation process can produce an isotropic concentration field around a nucleated particle. Moreover, it is possible for the internal oxide front to travel at such a velocity through the specimen that the rate of growth of a precipitate is of the same order as the rate of penetration of the internal oxidation front. In such a situation, a particular nucleus near the internal oxidation front would grow as a granular particle, exhausting all the Al plus O from one side but would continue to draw more material from the direction of the front. Then, the precipitation would lose its granular shape and would extend towards the front. Retention of a granular shape by the particle through volume or surface diffusion of Al and O atoms is slow relative to the growth process.

At this point, it is important to mention that the internal oxides in the Ni-Al alloys showed very particular morphologies, quite different from those of other alloy systems, such as Ni-Cr, Ni-V, Ni-Mo [20]. This is probably due to a very high directional precipitation in this alloy system. During exclusive internal oxidation, the reason for those rod-like morphologies is probably associated with the fact that, at the beginning of the internal oxidation process, a high population density of spherical particles can nucleate coherently with the matrix at random positions and start to grow in particular directions, probably in the direction of the internal oxide front. As growth continues, coherency is lost by the introduction of stress due to the high volume increase caused by the formation of Al_2_O_3_ particles. Then, the subsequent growth is controlled by the diffusional processes, but these will be conditioned to the amount of aluminum available in the alloy. As was observed for the Ni-1.18%Al alloy, the internal oxide precipitation resulted in spherical particles, fork-like triangles and acicular precipitates (Figure 3a–c). In this case, there was no sufficient aluminum available, and continuity of the precipitation through the internal oxidation zone was not possible.

When the solute content was increased (Ni-2.47%Al), formation of cylindrical rods in a cone-shape configuration (Figure 3d) continuous from the surface to the internal oxide/alloy front, was observed.

As the aluminum content was further increased (Ni-5.18% and Ni-8.67%Al), only cylindrical rods were produced. These grew through the internal oxidation zone, often in a cone-shaped configuration, where the thin part of the cones was situated at the surface, and the broad base of these cones was situated at the end of the internal oxidation front. Moreover, the separation among the cones was almost constant (Figure 3d). In this case, probably more aluminum is available, and hence, more nuclei can form at the beginning of the internal oxidation process, more elastic interaction develops, and the alignment of the precipitation increases until coherency is lost. Then, a diffusional process controls the subsequent growth in a cone-shape configuration (Figure 3d) instead of fork-like triangles, as was seen in Ni-1.18%Al (Figure 3b,c).

Diffusion of oxygen becomes more difficult through the metal matrix within the cones and along the internal oxide/metal interfaces on the inside of those cones, due to the high population density of precipitation. In effect, oxygen has difficulty in penetrating inside due to the cone configuration of rods. The subsequent growth of the internal oxidation front is probably enhanced by rapid diffusion of oxygen along the outside of the cone/metal interface, and at the end of the internal precipitation front, the number of precipitates within these cones is small. At this point, it is important to compare the cone-shape configuration formed in the Ni-Al alloys. In the Ni-2.47% Al alloy (Figure 3d) the cone-shape configuration consists of a number of cylindrical rods at a certain inclination with respect to the specimen surface and extending throughout the internal oxidation front. However, near to this front, there were gaps between precipitates. As the aluminum content is increased (Ni-8.67% Al, the gaps between the rod-precipitates are considerably fewer than in the Ni-2.47% Al case and now the cone-shape configuration can be considered to take the form approximately of an almost solid hollow cone. All these observations suggest that, when the bases of the cones touch each other, diffusion of oxygen inwards is reduced or blocked completely. Then oxygen attempts to diffuse laterally through any remaining gaps between the rods that make up the cone configuration. This blocking effect can allow diffusion of aluminum to the internal oxide front/metal interface from the bulk alloy. This aluminum reacts with oxygen and, if sufficient aluminum is available, an aluminum oxide healing layer can be formed and suppress any subsequent diffusion of oxygen into the alloy metal. As a matter of fact, in the Ni-8.67%Al alloy, a healing layer was observed along almost all the internal oxide front/metal interface after 20 h., at 1100 °C (Figure 5); in this case, diffusion on oxygen was observed and made some kind of shape.

The X-ray diffraction patterns (Ni-5.18% Al) in Figure 6 indicate that after 1 h of internal oxidation a small signal of NiAl_2_O_4_ becomes visible, and when increasing the internal oxidation exposure time (from 10 and 20 h), the signal becomes stronger. At this time, it is important to point out that the X-ray analysis was performed on the specimen surface without deep-etching, so the NiAl_2_O_4_ (spinel) signal is weak because nickel from the matrix masks the spinel signal. After deep-etching, the spinel signal becomes stronger. In some cases, it was observed that the Ni atoms diffused away into liquid Al from the solid/liquid interface due to the concentration gradient. During cooling, Ni atoms precipitated as Al_3_Ni in supersaturated liquid Al [21]; higher temperatures in the cutting zone promote thermally activated processes such as diffusion or plastic deformation [22].

In Figure 7, it can be seen that the diffraction patterns that are observed in several regions of the internal oxidation zone were different from each other, but they all form a pattern of points. Depending on the crystal orientation, it is possible to find superplasticity; this is obtained by Ni alloys. Some studies found cases of recrystallized grains. The pattern shown in Figure 7A allows to observe the substrates of single-crystal, next the image shows in Figure 7B a polycrystalline phase; that is a result of the transformation of the different metal orientations. Figure 7C,D has been attributed to differences in the nucleation rate as a result in differences in the sizes of the oxide cells. Figure 7E,F shows a martensite phase and transformation twins. In Figure 7E the TEM micrographs and the resulting oxide scale are observed.

## 5. Conclusions

In the Ni-Al alloys, several internal oxide morphologies were observed, depending upon aluminum concentration after oxidation in Ni/NiO Rhines pack at a given temperature. At low concentrations (Ni-1.18% Al), the internal oxide precipitates were a mixture of spherical particles, fork-like triangles and acicular precipitates, extending across the internal oxidation zone; these morphologies were observed at temperatures from 800 °C to 1100 °C.In the Ni-2.47%Al alloy, the internal oxides at 1000 and 1100 °C were continuous rod-like precipitates extending across the internal zone. At 800 °C, a rod-like was observed. For the Ni-5.18% Al and Ni-8.67% Al alloys, only rod-like precipitates were observed for all temperature-time conditions studied. The cylindrical, rod-like precipitates were mainly closely-spaced and oriented approximately perpendicular to the specimen surface.The cylindrical rods had grown through the internal oxidation zone in a cone-shaped configuration, where the thin part of the cone was situated at the surface, and the broad base of the cone was situated at the end of the internal oxidation front.The particles were Al_2_O_3_ in the vicinity of the internal oxide front, and NiAl_2_O_4_ was detected in the vicinity of the surface by ROI analyzed. Morphologies observed had a mixture of figures like triangles, spheres and cones related to oxidation zone. Using ROI allows to understand a little bit about oxidation.In superalloys, observations are related to the microstructure with results such as a vacancy diffusion and creep dislocations. In these studies, pattern analysis in Figure 6 shows that reactions depend on crystal orientation.

## Figures and Tables

**Figure 1 materials-13-00048-f001:**
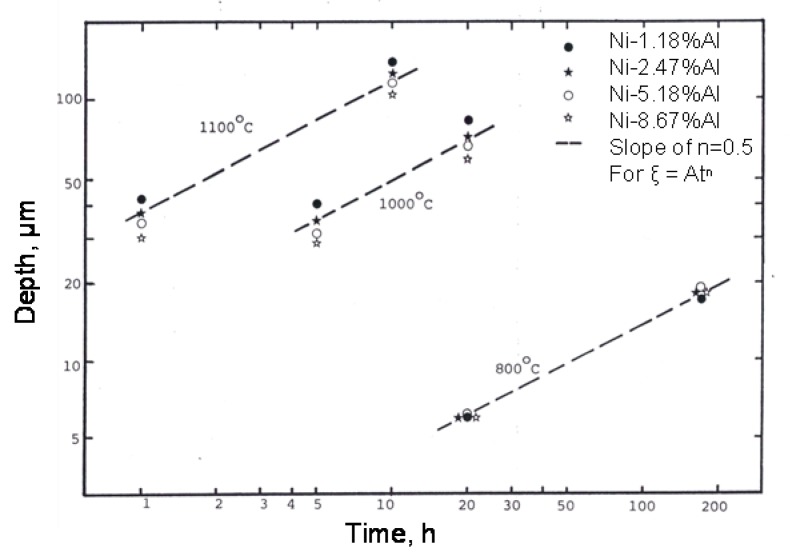
Uniform internal oxidation depth-time relationships in Ni-Al alloys Oxidized in Ni/NiO Rhines pack.

**Figure 2 materials-13-00048-f002:**
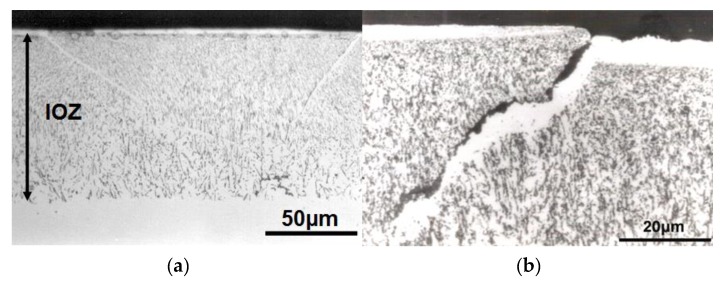
(**a**) Typical optical cross-section of the Ni-2.47% Al alloy oxidized in Ni/NiO Rhines pack at 1100 °C for 10 h. (**b**) Internal oxides appeared to be acicular.

**Figure 3 materials-13-00048-f003:**
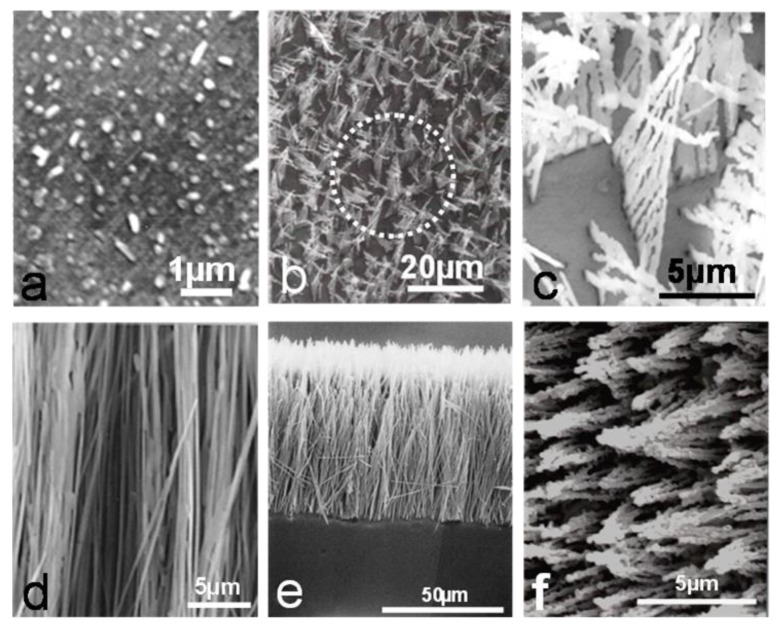
Some aspects of the deep-etching in Ni-Al alloys. (**a**) Ni-1.18%Al at 800 °C top view, (**b**) Ni-1.18% Al at 1100 °C general top view, (**c**) Ni-1.18% Al at 1100 °C zoom top view, (**d**) Ni-2.47% Al at 1000 °C top view, (**e**) Ni-5.18% Al at 1000 °C cross section view, (**f**) Ni-5.18% Al at 1100 °C cross section view.

**Figure 4 materials-13-00048-f004:**
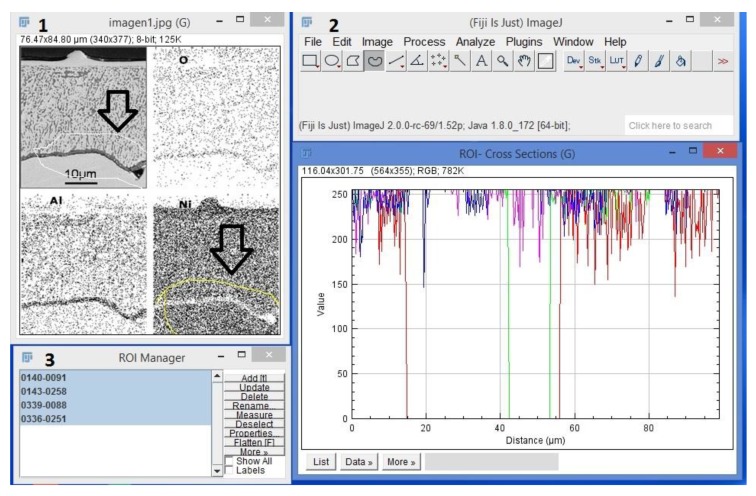
Multiples ROI selected on Fiji Software. (**1**): Region of Interest selected (**2**): Graphical result of ROI (**3**): ROI analysis.

**Figure 5 materials-13-00048-f005:**
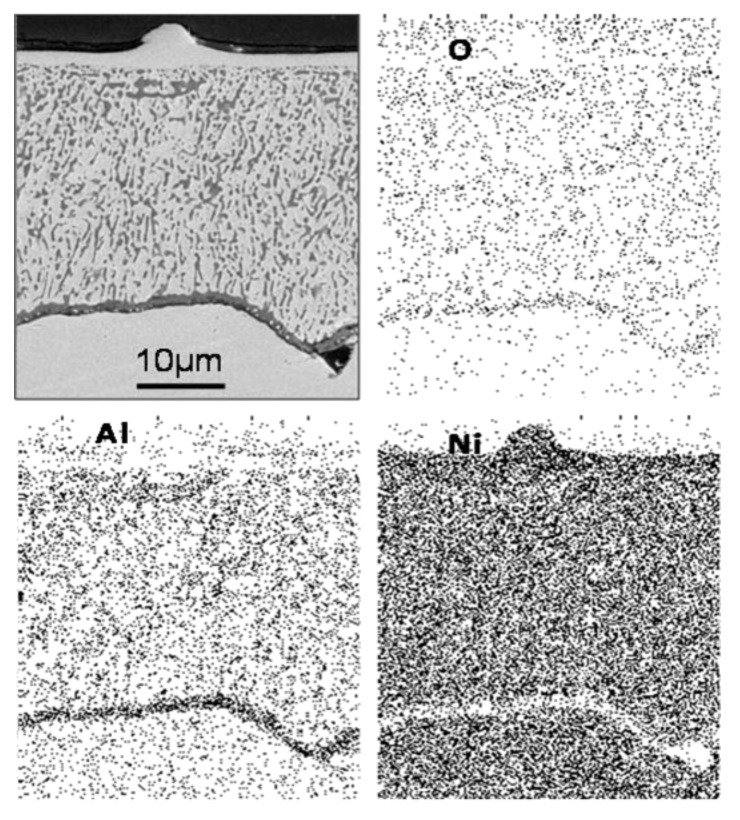
The number of cone-forming cylindrical rods increases with increasing solute concentration.

**Figure 6 materials-13-00048-f006:**
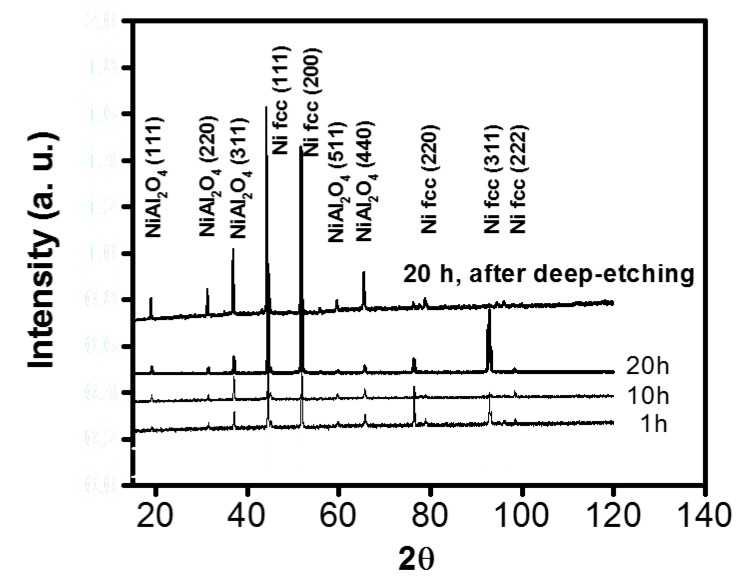
Ni-8.67% Al alloy oxidized at 1100 °C for 10 h showing (**a**) SEM view of aluminum oxide healing layer and X-ray Maps for (**b**) Al, (**c**) O and (**d**) Ni.

**Figure 7 materials-13-00048-f007:**
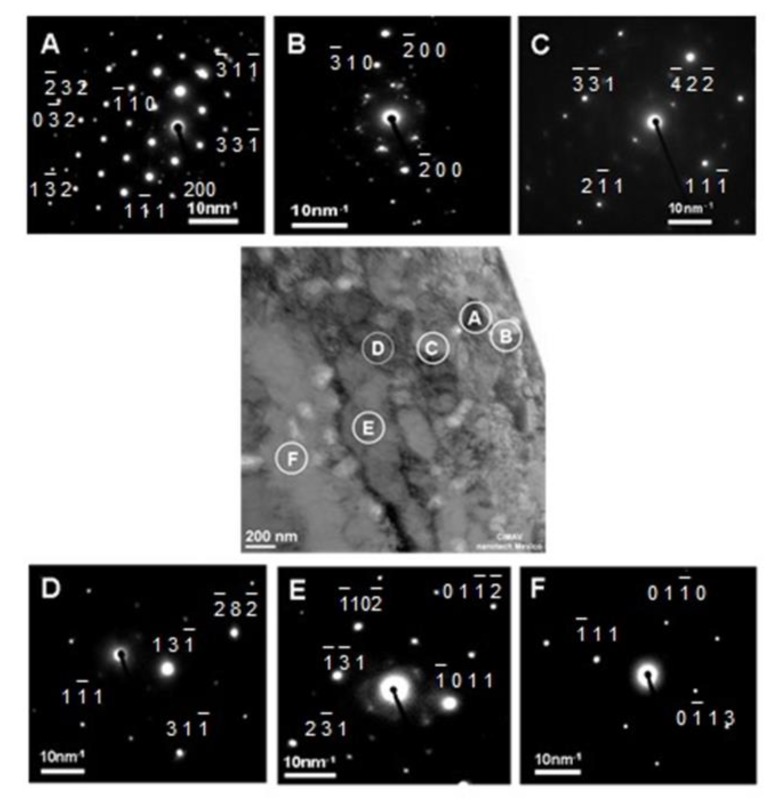
X-ray diffraction pattern for Ni-5.18 %Al alloy with and without deep-etching. (**A**): single-crystal (**B**): polycrystalline phase (**C**): nucleation rate (**D**): differences in the sizes of the oxide cells (**E**): martensite phase (**F**): transformation twins.

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
