# Peer review of "Estimating the Internal Oxidation of Ni-Al Alloys"

_materials, 2019, doi:10.3390/ma13010048_

Round 1

Reviewer 1 Report

Perhaps you can rearrange the figures, all of them are in a row. Extend the text and put them separated.

Is Fig 1 yours?. If not include the reference.

Fiji Software is commercial….include the symbol of copyright…what is REF 7, perhaps you can eliminate it….

References: I see some were missed, including some comments in the state of the art. Among those working in Ni alloys with Al in some amounts…

You addressed the topic:

“Ni-base super- alloys are complex engineering 29 alloys, primarily nickel with ten or more alloying elements, designed to operate at temperatures up 30 to 70% of their melting range while resisting mechanical and chemical degradation” but you do not give references as Journal of Manufacturing Processes 26, 44-56 showed several good ideas regarding secondary processes, in which oxides and other hard particles are key, extending in Journal of Manufacturing Processes 48, 44-50 state of the art must be complete, and your work is short in this aspect. Other journals Journal of Materials Engineering and Performance 25 (11), 5076-5086 delt with that as well

Materials: are they commercial of produced by you?. In metals. there are names and UNS numbers…please extend a little.

Conclusions: the findings are clear, but a little more explanation about the consequences in materials behavior would be welcome.

In short: make a better arrangement, extend the state of the art to the real target: Ni super alloys, and make figures clearer.

Author Response

Please see the attachment .pdf file

Reviewer 2 Report

This an interesting paper on the oxidation of Ni-Al alloys that, when published,  will be useful for super-alloy and other industrial applications. The microscopy and diffraction data is thorough, but several improvements are required before this work could be published:

Although the English in the paper is generally good, there are several minor edits that are required for clarity. For example, in line 22, the word "images" should be "image". In line 56, the first portion of the compound sentence beginning with "The morphology..." appears to be incomplete. In line 63, the word "and" in the phrase "reasonably and accurate" should be eliminated. In line 173, the word "growth" should be changed to "grow". In line 83, the analytical treatment described as "deep-etching" was used. The details of this methodology should be described (e.g. chemicals and conditions used, what is etched by the method and what is left behind) Figure captions for Figures 4, 4a, 5 and 6 need to be improved to provide more detail. For example, the caption for Figure 4 needs to provide an explanation that can stand alone. (I see only a one sentence description of the figure in the discussion beginning on line 210 and no mention in the results). I suggest re-numbering the figures so that Figure 4a becomes Figure 5, etc. In the current Figure 5 caption it says the results are for a sample oxidized at 1100C for 10 hours, but the figure also shows 1 hour, 20 hour and 20 hour after deep-etching results. The caption needs to be corrected and should be briefly described in the results. Each diffraction pattern and the sample should be mentioned in the Figure 6 caption and the Figure should be mentioned in the results.    (optional) The data in figure 1 plotting oxidation depth as a function of time is well behaved, but I would encourage the investigators to measure oxidation depth at a few intermediate times for at least 1 other alloy (if not more) so that the reader can be assured that there is truly a linear relationship between depth and log of time.  

Author Response

Please see the attachment.,pdf file

Round 2

Reviewer 1 Report

Paper and method are OK. My opinion is that the evaluation must be minor. ANYWAY THERE MUST BE MANDATORILY SOME CHANGES TO DO.

Always happen. Some results are good but the discussion is weaker.

Amazingly some references about the state of the art of Ni-Al are missed, paper is short. The GKN group was concerned about the subject and some ideas about state of the material was defined in Surface integrity and fatigue of non-conventional machined Alloy 718, Journal of Manufacturing Processes 48, 44-50.

Figure 4 a: it needs several changes. This is a direct screen from software.

GKN is one of the most concerned about Ni oxidation and its effects on fatigue and metallurgical state.

Include more references about Wretland, the idea is tricky because the papers are on manufacturing, but the way to deal with materials properties is key, they worked with Suarez et al. as well. My guess is that this third opinion about the paper. Please make it a little longer. Result are fruitful but Beranoagirre worked on the effect of the Al on titanium and other alloys as well. https://doi.org/10.1007/s00170-011-3812-6. Among others.
